# Left Atrial Thrombus in Atrial Fibrillation/Flutter Patients in Relation to Anticoagulation Strategy: LATTEE Registry

**DOI:** 10.3390/jcm11102705

**Published:** 2022-05-11

**Authors:** Agnieszka Kapłon-Cieślicka, Monika Gawałko, Monika Budnik, Beata Uziębło-Życzkowska, Paweł Krzesiński, Katarzyna Starzyk, Iwona Gorczyca-Głowacka, Ludmiła Daniłowicz-Szymanowicz, Damian Kaufmann, Maciej Wójcik, Robert Błaszczyk, Jarosław Hiczkiewicz, Katarzyna Łojewska, Katarzyna Mizia-Stec, Maciej T Wybraniec, Katarzyna Kosmalska, Marcin Fijałkowski, Anna Szymańska, Mirosław Dłużniewski, Maciej Haberka, Michał Kucio, Błażej Michalski, Karolina Kupczyńska, Anna Tomaszuk-Kazberuk, Katarzyna Wilk-Śledziewska, Renata Wachnicka-Truty, Marek Koziński, Paweł Burchardt, Piotr Scisło, Radosław Piątkowski, Janusz Kochanowski, Grzegorz Opolski, Marcin Grabowski

**Affiliations:** 1“Club 30”, Polish Cardiac Society, 02-097 Warsaw, Poland; agnieszka.kaplon@gmail.com (A.K.-C.); moni.budnik@gmail.com (M.B.); pkrzesinski@wim.mil.pl (P.K.); m.wojcik@umlub.pl (M.W.); kmiziastec@gmail.com (K.M.-S.); maciejwybraniec@gmail.com (M.T.W.); marcin.fijalkowski@gumed.edu.pl (M.F.); anna.szymanska@wum.edu.pl (A.S.); mhaberka@op.pl (M.H.); bwmichalski@gmail.com (B.M.); karolinakupczynska@gmail.com (K.K.); a.tomaszuk@poczta.fm (A.T.-K.); marekkozinski@wp.pl (M.K.); pab2@tlen.pl (P.B.); radekp1@gmail.com (R.P.); marcin.grabowski@wum.edu.pl (M.G.); 21st Department of Cardiology, Medical University of Warsaw, 02-097 Warsaw, Poland; piotr.scislo@gmail.com (P.S.); kochanowski.janusz@gmail.com (J.K.); grzegorz.opolski@gmail.com (G.O.); 3Institute of Pharmacology, West German Heart and Vascular Centre, University Duisburg-Essen, 45147 Essen, Germany; 4Department of Cardiology, Maastricht University Medical Centre and Cardiovascular Research Institute Maastricht, 6229 ER Maastricht, The Netherlands; 5Department of Cardiology and Internal Diseases, Military Institute of Medicine, 01-755 Warsaw, Poland; beata.zyczkowska@gazeta.pl; 61st Clinic of Cardiology and Electrotherapy, Świętokrzyskie Cardiology Centre, 25-558 Kielce, Poland; zikas@poczta.onet.pl (K.S.); iwona.gorczyca@interia.pl (I.G.-G.); 7Department of Cardiology and Electrotherapy, Medical University of Gdańsk, 80-210 Gdansk, Poland; ludmila.danilowicz-szymanowicz@gumed.edu.pl (L.D.-S.); d.kaufmann@gumed.edu.pl (D.K.); 8Department of Cardiology, Medical University of Lublin, 20-059 Lublin, Poland; robertblaszczyk1@wp.pl; 9Collegium Medicum, University of Zielona Góra, 65-417 Zielona Gora, Poland; jhiczkiewicz@uz.zgora.pl; 10Clinical Department of Cardiology, Nowa Sól Multidisciplinary Hospital, 67-100 Nowa Sol, Poland; katarzyna.lojewska@poczta.onet.pl; 111st Department of Cardiology, School of Medicine in Katowice, Medical University of Silesia, 40-055 Katowice, Poland; 12Department of Cardiology, St. Vincent Hospital, 81-348 Gdynia, Poland; katarzynakosmalska11@gmail.com; 131st Department of Cardiology, Medical University of Gdansk, 80-210 Gdansk, Poland; 14Department of Heart Diseases, Postgraduate Medical School, 00-002 Warsaw, Poland; miroslaw.dluzniewski@wum.edu.pl; 15Department of Cardiology, School of Health Sciences, Medical University of Silesia, 40-055 Katowice, Poland; michalek_k@poczta.fm; 16Department of Cardiology, Medical University of Lodz, 90-419 Lodz, Poland; 17Department of Cardiology, Medical University of Bialystok, 15-089 Bialystok, Poland; kat_wilk@wp.pl; 18Department of Cardiology and Internal Medicine, Medical University of Gdańsk, 80-210 Gdynia, Poland; kasiatr@mp.pl; 19Department of Hypertension, Angiology, and Internal Medicine, Poznan University of Medical Sciences, 61-701 Poznan, Poland

**Keywords:** thromboembolic risk, apixaban, dabigatran, rivaroxaban, transesophageal echocardiography

## Abstract

Background: Atrial fibrillation (AF) and flutter (AFl) increase the risk of thromboembolism. The aim of the study was to assess the prevalence of left atrial thrombus (LAT) in AF/AFl in relation to oral anticoagulation (OAC). Methods: LATTEE (NCT03591627) was a multicenter, prospective, observational study enrolling consecutive patients with AF/AFl referred for transesophageal echocardiography before cardioversion or ablation. Results: Of 3109 patients enrolled, 88% were on chronic, 1.5% on transient OAC and 10% without OAC. Of patients on chronic OAC, 39% received rivaroxaban, 30% dabigatran, 14% apixaban and 18% vitamin K antagonists (VKA). Patients on apixaban were oldest, had the worst renal function and were highest in both bleeding and thromboembolic risk, and more often received reduced doses. Prevalence of LAT was 8.0% (7.3% on chronic OAC vs. 15% without OAC; *p* < 0.01). In patients on VKA, prevalence of LAT was doubled compared to patients on non-VKA-OACs (NOACs) (13% vs. 6.0%; *p* < 0.01), even after propensity score weighting (13% vs. 7.5%; *p* < 0.01). Prevalence of LAT in patients on apixaban was higher (9.8%) than in those on rivaroxaban (5.7%) and dabigatran (4.7%; *p* < 0.01 for both comparisons), however, not after propensity score weighting. Conclusions: The prevalence of LAT in AF is non-negligible even on chronic OAC. The risk of LAT seems higher on VKA compared to NOAC, and similar between different NOACs.

## 1. Introduction

Atrial fibrillation (AF) and atrial flutter (AFl) increase the risk of thromboembolic events [1,2,3]. Current guidelines recommend initiation of oral anticoagulation (OAC) with vitamin K antagonists (VKAs) or, preferably, non-VKA-OACs (NOACs) in men and women with a CHA2DS2VASc score of ≥2 and ≥3, respectively [4]. Effective OAC is also recommended three weeks before elective AF cardioversion or catheter ablation with pre-procedural transesophageal echocardiography (TEE) as an alternative [4]. However, TEE before AF/AFl cardioversion or ablation is often performed even in anticoagulated patients, given that, in clinical practice, “effective” OAC may be difficult to achieve and/or verify.

The aim of the study was to assess the prevalence of LA thrombus in real-world AF/AFl patients referred for TEE before electrical cardioversion or catheter ablation in relation to the presence and the type of OAC.

## 2. Materials and Methods

### 2.1. Patient and Public Involvement

The Left Atrial Thrombus on Transesophageal Echocardiography (LATTEE) registry (NCT03591627) was a prospective, observational study enrolling consecutive patients with AF or AFl in whom TEE was performed before direct current cardioversion or catheter ablation, hospitalized in 13 cardiology departments (11 academic centers and two territorial departments) in Poland. Details on the study rationale and design have been reported elsewhere [5]. The patient recruitment process started in November 2018 in the coordinating centre and lasted 12 months since the beginning of the study in each participating centre or longer, i.e., until the inclusion of at least 200 patients at each participating centre (with the last patient enrolled in May 2020). Patients in whom TEE was performed several times during the study period were entered in the database under the same number.

The diagnosis of AF/AFl was made by attending physicians in accordance with the current guidelines [6]. Referral for TEE and performance of other diagnostic tests depended on the routine practice of a particular center and on the decision of an attending physician. Ten centers with electrophysiology teams performed TEE before catheter ablation in all AF/AFl patients and three centers in those questionable about adherence to OAC. Regarding non-emergency electrical cardioversion for AF/AFl, four centers performed TEE routinely in all patients, and nine centers performed TEE only in patients not pre-treated with OAC or in case of uncertainty of patients’ adherence to NOAC or, in patients on VKA, if the international normalized ratio (INR) was in non-therapeutic ranges.

The study was approved by the Ethics Committee of Medical University of Warsaw (AKBE/113/2018). Data were entered into the registry database anonymously. No additional tests or interventions, apart from those planned by the attending physicians, were performed. Thus, the ethics committee waived the requirement of obtaining informed consent from the patients.

### 2.2. Data Collection

Data was gathered prospectively and included demographics, medical history, current pharmacotherapy, results of routine laboratory tests (on hospital admission), and TEE results in all patients. Chronic OAC was defined as OAC treatment for at least three weeks before the procedure. Transient OAC was defined as OAC prescribed for less than three weeks, usually a few days before the procedure. Glomerular filtration rate (GFR) was estimated using the Cockcroft-Gault formula: ([140-age] × weight in [kilograms/serum creatinine] in [mg/d]× 72) × 0.85 (if female). The primary outcome measure was the prevalence of LA thrombus on TEE. Apart from LA thrombus, data on spontaneous echo contrast (SEC), including dense SEC (grades 3–4 in the scoring system [7]) were also gathered. LA appendage emptying velocity was measured 1 cm below the orifice of the appendage. All TEE studies were performed by certified echocardiographers.

### 2.3. Statistical Analysis

All continuous variables were tested for normality with the Kolmogorov-Smirnov test. Nonparametric variables were expressed as median and interquartile range [IQR], and categorical variables as counts (*n*) with percentages (%). Fisher’s exact test (two group comparisons) or Chi-square tests (three group comparisons) were used to compare categorical variables. Differences in continuous parameters were compared using a Mann-Whitney U test (two group comparisons) and Kruskal-Wallis test (three groups comparisons) in case of nonparametric variables and unpaired *t*-test (two group comparison) or ANOVA (three groups comparison) in the case of parametric variables. To compare the risk of LA thrombus in patients on different OAC regimens, univariate (unadjusted) and multivariable (adjusted) logistic regression analyses were performed. In multivariable analyses, the effect of OAC strategy on the risk of LA thrombus was adjusted for age, sex, the presence of heart failure, hypertension, diabetes, vascular disease, and previous ischemic stroke, transient ischemic attack (TIA) or systemic embolism (variables included in the CHA_2_DS_2_-VASc score), and, additionally, for AF/AFl type (non-paroxysmal vs. paroxysmal), GFR and concomitant antiplatelet treatment. While comparing the prevalence of LA thrombus between different OAC strategies, to adjust for potential confounding due to baseline imbalances in study covariates while preserving sample size, we used propensity score weighting. With this method, the propensity score was used to generate patient-specific stabilized weights that control for covariate imbalances. Covariates used for propensity score weighting included age ≥65 years, sex, heart failure, hypertension, vascular disease, diabetes, ischemic stroke/TIA/systemic embolism (variables included in the CHA_2_DS_2_-VASc score), as well as GFR < 50 mL/min and LA appendage emptying velocity. Covariate balance between the weighted cohorts was assessed using standardized mean differences. A standardized difference of 0.05 or more indicates a negligible difference between groups. The distributions of propensity scores and stabilized weights were inspected for outliers. A two-sided *p* value of 0.05 was considered statistically significant. For database management and statistical analysis, we used SAS 14.1 (SAS Institute Inc., Cary, NC, USA).

## 3. Results

### 3.1. Population

Overall, 3109 patients were enrolled in the LATTEE registry: 2689 (86%) in academic and 420 (14%) in territorial centers. Median age was 67 (59–73) years, 37% were women, 85% had AF, 12% AFl, and 3.1% both AF and AFl. Median CHA_2_DS_2_-VASc score was 3 (2–4). In 52%, TEE was performed before cardioversion. Detailed characteristics of the LATTEE population are presented in Appendix A.

### 3.2. Oral Anticoagulation

Data on OAC therapy (none, transient or chronic) was available for 3107 (99.9%) patients, with 88% on chronic OAC, 1.5% on transient OAC, and 10% without OAC (Figure 1A). Of patients on chronic OAC, 18% received VKA and 82% NOAC (17% of the latter received reduced doses; Figure 1B). Proportions of patients on chronic OAC were similar regardless of class of recommendation (89% in patients with class I indications to chronic OAC, 88% with class IIa indications and 85% with no indications; Figure 1C). Compared to patients on chronic OAC, those with no previous OAC were more often referred for cardioversion and less often for ablation. They had similar median age and CHA_2_DS_2_-VASc score, but they more often had a history of bleeding, had slightly lower GFR and were more often treated with antiplatelets. They received periprocedural heparin significantly more often than those on chronic OAC (35% vs. 4%; Appendix A).

### 3.3. Left Atrial Thrombus

Overall, the prevalence of LA thrombus on TEE was 8.0% and was lowest in patients on chronic OAC (7.3%) and highest in those with no OAC (15%; *p* < 0.01; Figure 2A). 96% of LA thrombi were localized in its appendage. The prevalence of LA thrombus in patients with both AF and AFl, only AF and only AFl episodes, was 4.2%, 8.2% and 8.2% (*p* = 0.37), respectively. Compared to patients with no thrombus, those with LA thrombus were older, more often had non-paroxysmal AF, had more comorbidities with higher median CHA_2_DS_2_-VASc score (4 (3–5) vs. 3 (2–4); *p* < 0.01), were less often treated with OAC, but were treated with antiplatelets more often and more often received periprocedural heparin (Appendix A). Still, most LA thrombi occurred in patients on chronic OAC (80%) and with class I indications to OAC (93%) (Appendix A). The prevalence of LA thrombus in anticoagulated patients with class I indications to OAC was 8.6%, with class IIa indications 2.4%, and with no indications 0.9% (*p* < 0.01; Appendix A).

### 3.4. VKA vs. NOAC

In patients on VKA, the prevalence of LA thrombus was twice as high as in patients on NOAC (13% vs. 6.0%; *p* < 0.01; Figure 2B). Compared to patients on NOAC, those on VKA were older, more often had non-paroxysmal AF and, unsurprisingly, mechanical valve prostheses and had more comorbidities with slightly higher CHA_2_DS_2_-VASc score (with a median score of 3 for both VKA and NOAC), and with somewhat worse renal function (Table 1). However, even after multiple adjustments for potential confounders, the risk of LA thrombus remained twice as high with VKA as with NOAC (Figure 3). In propensity score weighting, the prevalence of LA thrombus in 491 patients on VKA was 13% and in 491 matched patients on NOAC it was 7.5% (*p* < 0.01; Appendix A).

### 3.5. Apixaban vs. Rivaroxaban vs. Dabigatran

Of patients on chronic OAC, 39% received rivaroxaban, 30% dabigatran and 14% apixaban (Figure 1A). Patients treated with apixaban more often received reduced doses (29%) compared to patients on rivaroxaban (14%) and dabigatran (17%; *p* < 0.01 for both comparisons; Figure 1B). Patients on apixaban were older, more often had non-paroxysmal AF, had more comorbidities with higher median CHA_2_DS_2_-VASc score (4 [2,3,4,5] for apixaban vs. 3 [2,3,4] for rivaroxaban and 3 [1,2,3,4] for dabigatran; *p* < 0.01), had lower GFR, and more often a history of bleeding and anemia, and were more often treated with antiplatelets compared to patients on other NOACs (Table 1).

The prevalence of LA thrombus in patients on apixaban was twice as high (9.8%) as in those on rivaroxaban (5.7%) and dabigatran (4.7%; *p* < 0.01 for both comparisons; Figure 2B) and this difference remained in patients on standard NOAC doses (8.4% for apixaban vs. 4.3% for rivaroxaban and 4.0% for dabigatran; Figure 2C). Thus, the unadjusted risk of LA thrombus for apixaban was doubled compared to other NOAC (Figure 3A). However, after multiple adjustments for potential confounders (variables included in the CHA_2_DS_2_-VASc score, AF type, renal function [GFR] and antiplatelet treatment), there was no difference in the risk of LA thrombus between apixaban and other NOACs (both in all patients on NOAC and in those on standard NOAC doses; Figure 3C,D). In propensity score weighting, the prevalence of LA thrombus in patients treated with standard doses of dabigatran, rivaroxaban and apixaban was comparable (Appendix A).

## 4. Discussion

The major findings of our study are as follows. First, in a real-world population of AF/AFl patients referred for TEE before cardioversion or ablation, the prevalence of LA thrombus was 7.3% despite chronic OAC. Second, the risk of LA thrombus seems higher on VKA compared to NOAC. Third, apixaban is more often prescribed in patients with higher bleeding risk and therefore also with higher thromboembolic risk, which may account for the higher prevalence of LA thrombus in apixaban-treated patients. However, after adjusting for clinical characteristics, the risk of LA thrombus for apixaban seems similar to that for rivaroxaban and dabigatran.

Similar to our study, in a recent meta-analysis of observational studies, the prevalence of LA thrombus in 9772 anticoagulated patients undergoing TEE before cardioversion or catheter ablation was 6.7%, with higher prevalence in patients on VKA (12% on VKA vs. 4.7% on NOAC) [8]. In meta-analyses of randomized controlled trials compared to VKA, NOACs were shown to reduce all-cause mortality, as well as the risk of stroke or systemic embolism; the latter driven mainly by a ~50% reduction in hemorrhagic stroke [9]. In terms of prevention of ischemic stroke, NOACs have demonstrated similar efficacy to VKA, with only a trend towards benefit with NOACs [10]. Lower relative efficacy of VKA in observational studies reporting on the prevalence of LA thrombus on TEE [8,11] could be explained by lower time in therapeutic range (TTR) in real-world patients [12] compared to TTR achieved in patients enrolled in clinical trials [13], and could imply even more benefit with NOACs compared to VKA in real-world AF population. Still, not every LA thrombus results in ischemic stroke and, thus, the higher prevalence of LA thrombus under VKA treatment might not necessarily translate into a higher risk of thromboembolic events.

All NOACs seem to provide comparable protection from thromboembolic events, while apixaban consistently demonstrated lower bleeding risk compared to rivaroxaban and dabigatran [14]. Even at the standard dose, apixaban seems to convey the lowest risk of major bleeding [15]. Of all NOACs, apixaban is eliminated by the kidney to the least extent, therefore, contrary to other NOACs, it can be used at a standard dose in patients with GFR between 30 and 49 mL/min, unless other indications to dose reduction are present [16]. Although neither the 2020 guidelines of the European Society of Cardiology [4] nor the 2021 European Heart Rhythm Association practical guide on the use of NOACs [16] provide clear recommendations on the choice of NOAC in specific clinical situations, in our real-life AF/AFl population, we observed a preference for apixaban in patients with the highest bleeding risk and the worst renal function (and a preference for rivaroxaban and dabigatran in those with the lowest bleeding risk, with VKA prescribed in the “intermediate” group): patients on apixaban were the oldest (median age: 71 years vs. 68 years on VKA vs. ~66 years on rivaroxaban and dabigatran), more often had a history of previous bleeding (7.0% vs. 3.5% on VKA and ~3% on rivaroxaban and dabigatran) or anemia (19% vs. 22% on VKA vs. 12% on rivaroxaban and 15% on dabigatran) and more often had GFR < 50 mL/min (18% vs. 15% on VKA vs. ~8% on rivaroxaban and dabigatran). At the same time, patients on apixaban had the highest thromboembolic risk determined by both the highest median CHA_2_DS_2_-VASc score (4 vs. 3 on VKA, rivaroxaban and dabigatran) and the worst renal function [17,18]. This, together with more frequent prescription of reduced doses, could account for the higher prevalence of LA thrombus in apixaban-treated patients compared to rivaroxaban- and dabigatran-treated patients. Importantly, both in multivariable logistic regression and propensity score weighting, the risk of LA thrombus for apixaban was comparable to other NOACs. On the other hand, for VKA, the risk of LA thrombus remained significantly higher compared to NOAC even after multiple adjustments and in propensity score weighting.

Current guidelines recommend TEE before AF cardioversion or catheter ablation as an alternative to a 3-week course of OAC [4]. In our study, the prevalence of LA thrombus on chronic OAC was 8.6% in patients with class I indications to OAC and over 3 times lower (2.4%) in those with class IIa indications. In patients with class I indications, the prevalence of LA thrombus was high both on VKA (13%) and on NOAC (7.6%), whereas in those with class IIa indications it was still high on VKA (11%), but much lower on NOAC (1.2%, *p* < 0.01). Thus, it may be suggested that patients with high thromboembolic risk (class I indications to OAC) should not be exempted from TEE before cardioversion or catheter ablation even after three weeks of OAC treatment [19]. In patients with class IIa indications to OAC, TEE before cardioversion might be reasonable in patients treated with VKA if no evidence for effective TTR is available. Patients with no indications to OAC developed LA thrombus only occasionally (three patients, including two on chronic OAC).

### Limitations

Our study was an observational one, and therefore results regarding efficacy of different OAC regiments should be viewed with some care. Nevertheless, propensity score weighting and multivariable logistic regression, adjusted for potential confounders, brought similar results with respect to the efficacy of VKA vs. NOAC and of different NOACs. The registry-based character of our study has its limitations, but also advantages, providing real-world estimates of the factual prevalence of LA thrombus during OAC treatment. Nine participating centers did not perform TEE routinely in all patients referred for cardioversion, only in those with doubts regarding adherence to NOAC or effectiveness of OAC, which might have led to some selection bias. Furthermore, most (86%) patients were enrolled in academic centers, which also might have led to some selection bias. Data on the indications to electrical cardioversion (urgent vs. elective cardioversion, first vs. recurrent episode of AF) were not collected in the registry. Next, diagnostic tests were performed at the discretion of attending physicians, and thus, some data are missing for some patients. Therefore, in tables, we have given the number of patients for whom a given parameter was available. Data on INR before index hospitalization were missing in one third of VKA-treated patients, and assessment of TTR was not possible. On the other hand, the study reflected real-world patients on VKA, who sometimes fail to adequately control INR. Finally, the primary endpoint of our study was the presence of LA thrombus on TEE, and not ischemic stroke. Not all ischemic strokes in AF result from LA thrombi, and not all LA thrombi lead to ischemic stroke. However, thrombus formation in LA is considered the primary mechanism responsible for thromboembolic events in patients with AF; thus, the design of our study seems appropriate to address thromboembolic risk in AF/AFl [1,2,3].

## 5. Conclusions

Prevalence of LA thrombus in AF is non-negligible even on chronic OAC, therefore TEE should be considered, if feasible, before cardioversion/ablation in all patients with high thromboembolic risk (class I indications to OAC) irrespective of OAC treatment. In real-world AF/AFl population, the risk of LA thrombus seems higher on VKA compared to NOAC, and similar between different NOACs.

## Figures and Tables

**Figure 1 jcm-11-02705-f001:**
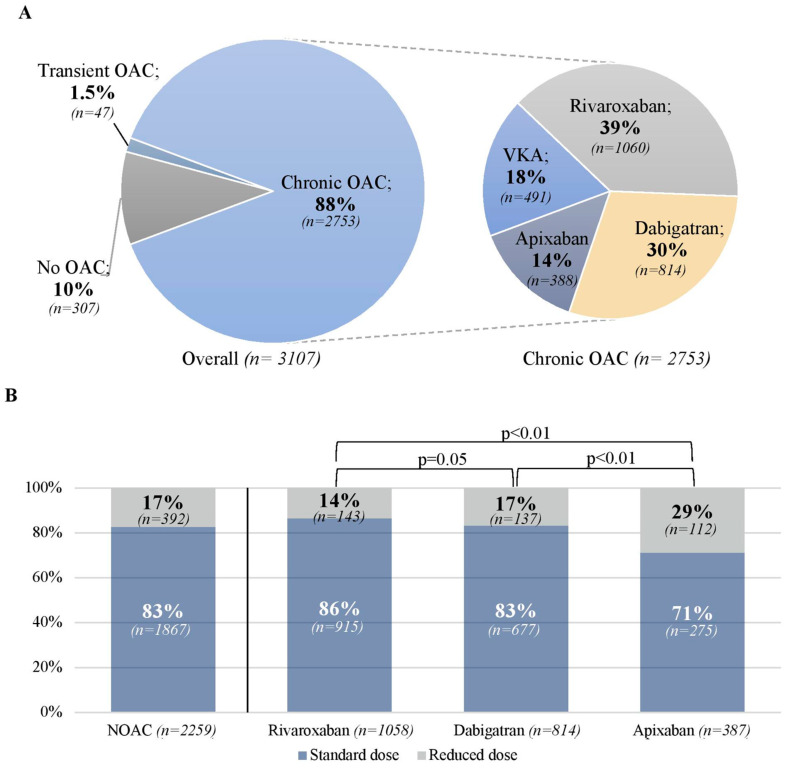
Oral anticoagulation in patients with atrial fibrillation/flutter undergoing transesophageal echocardiography before cardioversion or ablation. (**A**): Presence and type of oral anticoagulation (*n* = 3107). (**B**): Dosing of non-vitamin K oral anticoagulants (NOAC) (*n* = 2259). (**C**): Oral anticoagulation in relation to indications to chronic anticoagulation (*n* = 3087). Legend: No indications to chronic anticoagulation: CHA_2_DS_2_-VASc 0 (if male) and 1 (if female). Class IIa indications to chronic anticoagulation: CHA_2_DS_2_-VASc 1 (if male) and 2 (if female). Class I indications to chronic anticoagulation: CHA_2_DS_2_-VASc ≥ 2 (if male) and ≥3 (if female) or moderate-severe mitral stenosis or mechanical valve prosthesis; Reduced doses for rivaroxaban, dabigatran and apixaban were 15 mg o.d., 110 mg b.i.d. and 2.5 mg b.i.d., respectively. Only one patient in the rivaroxaban group received a reduced dose of 10 mg o.d. Only *p*-values of 0.05 or lower were shown for group comparisons; Abbreviations: NOAC, non-voitamin K oral anticoagulant; OAC, oral anicoagulation; VKA, vitamin K antagonist.

**Figure 2 jcm-11-02705-f002:**
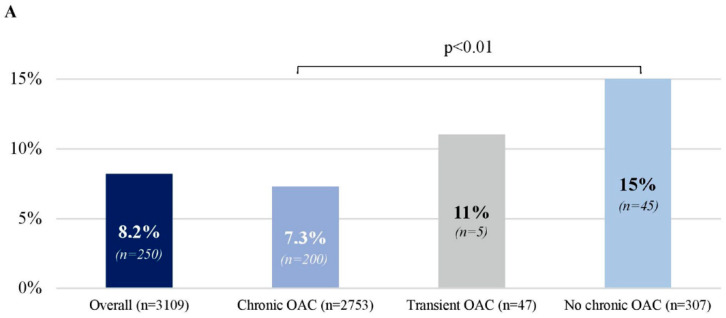
Prevalence of left atrial thrombus in patients with atrial fibrillation/flutter undergoing transesophageal echocardiography before cardioversion or ablation. (**A**): Overall (*n* = 3109). (**B**): In patients on chronic oral anticoagulation (*n* = 2753). (**C**): In patients on standard doses of non-vitamin K oral anticoagulants (NOAC) (*n* = 1867). Legend: Only *p*-values of 0.05 or lower were shown for group comparisons. Abbreviations: NOAC, non-voitamin K oral anticoagulant; OAC, oral anicoagulation; VKA, vitamin K antagonist.

**Figure 3 jcm-11-02705-f003:**
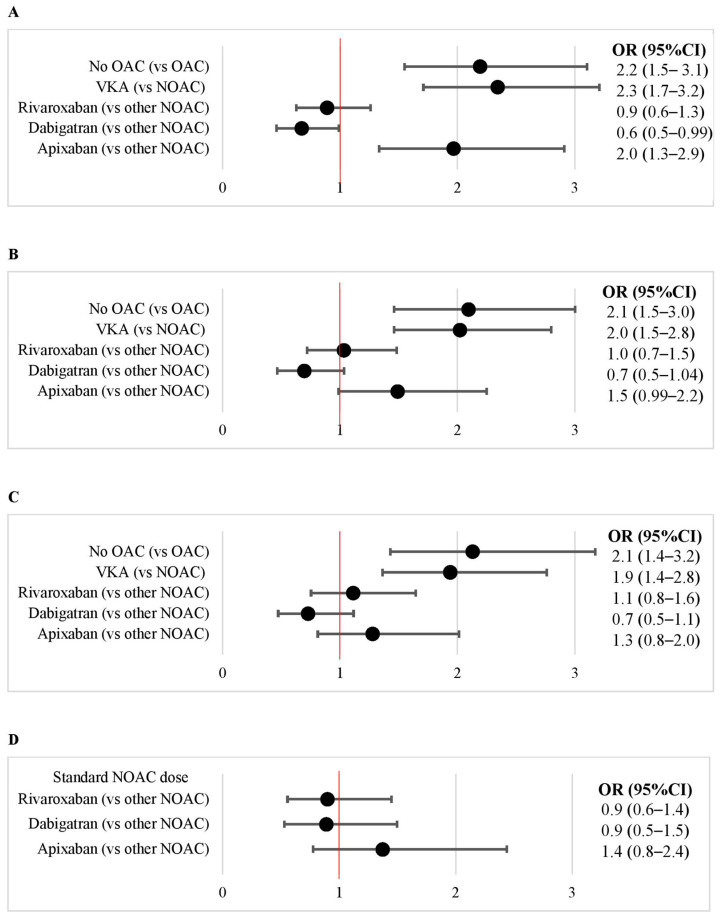
Risk of left atrial thrombus in patients with atrial fibrillation/flutter in relation to oral anticoagulation. (**A**): Unadjusted. (**B**): Adjusted for age, sex, heart failure, hypertension, diabetes, vascular disease, and previous ischemic stroke/TIA/systemic embolism. (**C**): Adjusted for age, sex, heart failure, hypertension, diabetes, vascular disease, previous ischemic stroke/TIA/systemic embolism, atrial fibrillation/flutter type (non-paroxysmal vs. paroxysmal), glomerular filtration rate, and antiplatelets. (**D**): In patients on standard doses of NOACs. Adjusted for age, sex, heart failure, hypertension, diabetes, vascular disease, previous ischemic stroke/TIA/systemic embolism, atrial fibrillation/flutter type (non-paroxysmal vs. paroxysmal), glomerular ejection fraction, and antiplatelets. Abbreviations: CI, coincidence interval; NOAC, non-voitamin K oral anticoagulant; OAC, oral anicoagulation; OR, odds ratio; VKA, vitamin K antagonist.

**Table 1 jcm-11-02705-t001:** Baseline characteristics of patients on chronic oral anticoagulation in relation to type of anticoagulation.

Variable	VKA (*n* = 491)	NOAC (*n* = 2262)	*p* ^1^	NOAC (*n* = 2262)			*p* ^2^
				Rivaroxaban (*n* = 1060)	Dabigatran (*n* = 814)	Apixaban (*n* = 388)	
Demographics							
Age (years)	68 [62–74] *n* = 490	66 [59–73] *n* = 2259	<0.01	66 [58–73] *n* = 1059	65 [58–72] *n* = 812	71 [63–77] *n* = 388	<0.01
Age ≥ 75 years	108/490 (22%)	471/2259 (21%)	0.58	200/1059 (19%)	146/812 (18%)	125/388 (32%)	<0.01
Female sex	196/491 (40%)	826/2262 (37%)	0.16	380/1060 (36%)	281/814 (35%)	165/388 (43%)	0.02
BMI (kg/m^2^)	29 [26–33] *n* = 463	29 [26–33] *n* = 2107	0.55	29 [26–33] *n* = 983	29 [26–33] *n* = 767	29 [26–32] *n* = 357	0.26
Indications for TEE							
Direct current cardioversion for AF/AFl	282/475 (59%)	1055/2236 (47%)	<0.01	501/1052 (48%)	321/804 (40%)	233/380 (61%)	<0.01
AF/AFl ablation	193/475 (41%)	1181/2236 (53%)	<0.01	551/1052 (52%)	483/804 (60%)	147/380 (39%)	<0.01
AF/AFl type							
AF	428/491 (87%)	2025/2262 (90%)	0.13	956/1060 (90%)	737/814 (91%)	332/388 (86%)	0.02
AFl	80/491 (16%)	309/2262 (14%)	0.13	137/1060 (13%)	102/814 (13%)	70/388 (18%)	0.02
AF/AFl paroxysmal	158/487 (32%)	969/2259 (43%)	<0.01	458/1059 (43%)	380/813 (47%)	131/387 (34%)	<0.01
AF/AFl persistent	262/487 (54%)	1107/2259 (49%)	0.06	528/1059 (50%)	360/813 (44%)	219/387 (57%)	<0.01
AF/AFl long-standing persistent	67/487 (14%)	183/2259 (8.1%)	<0.01	73/1059 (6.9%)	73/813 (9.0%)	37/387 (9.6%)	0.13
Comorbidities							
Hypertension	385/490 (79%)	1736/2261 (77%)	0.41	811/1060 (77%)	618/814 (76%)	307/387 (79%)	0.41
Heart failure	254/491 (52%)	902/2253 (40%)	<0.01	410/1057 (39%)	306/809 (38%)	186/387 (48%)	<0.01
Mechanical valve prosthesis	68/491 (14%)	3/2258 (0.1%)	<0.01	1/1060 (0.1%)	1/811 (0.1%)	1/387 (0.3%)	0.75
Biological valve prosthesis (including TAVI)	24/491 (4.9%)	24/2258 (1.1%)	<0.01	8/1060 (0.8%)	10/820 (1.2%)	6/387 (1.5%)	0.39
Vascular disease	193/490 (39%)	755/2261 (33%)	0.01	333/1060 (31%)	265/814 (33%)	157/387 (41%)	<0.01
Previous stroke	40/490 (8.2%)	171/2261 (7.6%)	0.64	69/1060 (6.5%)	63/814 (7.7%)	39/387 (10%)	0.07
Ischemic stroke/TIA/systemic embolism	56/488 (11%)	229/2258 (10%)	0.37	91/1060 (8.6%)	87/812 (11%)	51/386 (13%)	0.03
Previous bleeding	17/490 (3.5%)	82/2261 (3.6%)	1.00	29/1060 (2.7%)	26/814 (3.2%)	27/387 (7.0%)	<0.01
Diabetes mellitus	141/490 (29%)	538/2261 (24%)	0.02	236/1060 (22%)	194/814 (24%)	108/387 (28%)	0.08
GFR (mL/min)	80 [60–100] *n* = 439	82 [64–103] *n* = 2031	0.049	84 [67–106] *n* = 946	85 [66–105] *n* = 734	73 [57–90] *n* = 351	<0.01
GFR < 50 mL/min	65/439 (15%)	200/2031 (9.9%)	<0.01	76/946 (8.0%)	60/734 (8.2%)	64/351 (18%)	<0.01
COPD	31/490 (6.3%)	105/2261 (4.6%)	0.13	37/1060 (3.5%)	44/814 (5.4%)	24/387 (6.2%)	0.04
Anemia ^3^	102/472 (22%)	312/2189 (14%)	<0.01	123/1028 (12%)	116/785 (15%)	73/376 (19%)	<0.01
Thromboembolic risk and indications to chronic OAC							
CHA_2_DS_2_-VASc score	3 [2–5] *n* = 487	3 [2–4] *n* = 2246	<0.01	3 [2–4] *n* = 1056	3 [1–4] *n* = 805	4 [2–5] *n* = 385	<0.01
Class I indications to OAC ^4^	414/488 (85%)	1632/2246 (73%)	<0.01	767/1056 (73%)	550/805 (68%)	315/385 (82%)	<0.01
-moderate/severe MS or mechanical valve prosthesis	77/488 (16%)	10/2246 (0.4%)	<0.01	5/1056 (0.5%)	2/805 (0.2%)	3/385 (0.8%)	0.43
Class IIa indications ^5^	53/488 (11%)	408/2246 (18%)	<0.01	185/1056 (18%)	175/805 (22%)	48/385 (12%)	<0.01
No indications to chronic OAC ^6^	21/488 (4.3%)	206/2246 (9.2%)	<0.01	104/1056(9.9%)	80/805 (9.9%)	22/385 (5.7%)	0.04
International normalized ratio (INR) for patients on VKA							
Data on INR during hospitalization	473/491 (96%)	non-applicable					
INR 2–3 at hospital admission	197/473 (42%)						
Data on INR before hospitalization	332/491 (68%)						
INR 2–3 before hospitalization	178/332 (46%)						
Antithrombotic therapy							
Heparin (periprocedural):	32/489 (6.5%)	77/2255 (3.4%)	<0.01	38/1056 (3.6%)	27/812 (3.3%)	12/387 (3.1%)	0.89
-heparin ≥ 2 days	14/489 (2.9%)	9/2253 (0.4%)	<0.01	4/1056 (0.4%)	3/811 (0.4%)	2/386 (0.5%)	0.91
Antiplatelets	69/491 (14%)	169/2262 (7.5%)	<0.01	66/1060 (6.2%)	60/814 (7.4%)	43/388 (11%)	<0.01
Transesophageal echocardiography							
Left atrial appendage emptying velocity (cm/s)	30 [20–44] *n* = 419	40 [28–55] *n* = 1917	<0.01	40 [28–55] *n* = 897	41 [30–60] *n* = 695	36 [24–50] *n* = 325	<0.01
SEC	165/485 (34%)	523/2236 (23%)	<0.01	250/1048 (24%)	171/805 (21%)	102/383 (27%)	0.11
Left atrial thrombus	64/491 (13%)	136/2262 (6.0%)	<0.01	60/1060 (5.7%)	38/814 (4.7%)	38/388 (9.8%)	<0.01

Legend: ^1^
*p* value for the difference between patients on VKA vs. no NOAC; ^2^
*p* value for difference between rivaroxaban, dabigatran and apixaban; ^3^ hemoglobin < 12 g/dL for female < 13 g/dL for male; ^4^ CHA_2_DS_2_-VASc score ≥ 2 for men and ≥3 for women or moderate-severe MS or mechanical valve prosthesis; ^5^ CHA_2_DS_2_-VASc score 1 for men and 2 for women; ^6^ CHA_2_DS_2_-VASc score 0 for men and 1 for women. Abbreviations: AF, atrial fibrillation; Afl, atrial flutter; BMI, body mass index; COP, chronic obstructive pulmonary disease; GFR, glomerular filtration rate; OAC oral anticoagulation; NOAC, non-vitamin K oral anticoagulant; SEC, spontaneous echo contrast; VKA, vitamin K antagonist.

## Data Availability

Not applicable.

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
