# Peer review of "Left Atrial Thrombus in Atrial Fibrillation/Flutter Patients in Relation to Anticoagulation Strategy: LATTEE Registry"

_jcm, 2022, doi:10.3390/jcm11102705_

Round 1

Reviewer 1 Report

This prospective observational research study provides additional insight into the prevalence of left atrial thrombus in patients with high risk of thromboembolic events under chronic anticoagulation therapy. There are some questions to be answered:

  1. Did the authors perform a sensitivity analysis according to clinical centre and physician?
  2. Although relative risk and odds ratio should be similar in this setting, with logistic regression it is more appropriate to mention odds instead of relative risk.
  3. Did the authors consider differences between valvular and non-valvular atrial fibrillation?
  4. Were there any differences in outcomes between AF and flutter patients?
  5. The reference regarding this sentence is missing: “in a recent meta-analysis of observational studies, the prevalence of LA thrombus in 9772 anticoagulated patients undergoing TEE before cardioversion or catheter ablation was 6.7%, with higher prevalence in patients on VKA (12% on VKA vs. 4.7% on NOAC).”
  6. The authors mention Bonferroni correction, although in the statistical analysis and tables there is no mention of the Bonferroni-corrected p value.

Author Response

Reviewer 1

Comments and Suggestions for Authors

This prospective observational research study provides additional insight into the prevalence of left atrial thrombus in patients with high risk of thromboembolic events under chronic anticoagulation therapy. There are some questions to be answered:

  1. Did the authors perform a sensitivity analysis according to clinical centre and physician?

Sensitivity analysis was not performed.

  1. Although relative risk and odds ratio should be similar in this setting, with logistic regression it is more appropriate to mention odds instead of relative risk.

We apologise for this mistake in “Statistical analysis” section. We removed the word “relative” (page 3, line 112), given we provided the results as odds ratio, as is shown in Figure 3.

  1. Did the authors consider differences between valvular and non-valvular atrial fibrillation?

There was no difference in left atrial thrombus prevalence regarding mechanical valve prosthesis (3.6% in patients with left atrial thrombus vs 2.2% in those without left atrial thrombus, P=0.19), and biological valve prosthesis (2.8% vs 2.2%, P=0.20; Table S2; supplementary material online). The detailed comparison of impact of valvular heart disease in patients with atrial fibrillation is a topic of another study from the LATTEE registry, therefore we did not expand this topic in this article.

  1. Were there any differences in outcomes between AF and flutter patients?

We included that information as follows:

“The prevalence of LA thrombus in patients with both AF and AFl, only AF and only AFl episodes, was 4.2%, 8.2% and 8.2% (p=0.37), respectively”. (page 5, lines 167-169)

  1. The reference regarding this sentence is missing: “in a recent meta-analysis of observational studies, the prevalence of LA thrombus in 9772 anticoagulated patients undergoing TEE before cardioversion or catheter ablation was 6.7%, with higher prevalence in patients on VKA (12% on VKA vs. 4.7% on NOAC).”

We included the reference #8 (page 10, line 243).

  1. Noubiap, J.J., et al., Atrial thrombus detection on transoesophageal echocardiography in patients with atrial fibrillation undergoing cardioversion or catheter ablation: A pooled analysis of rates and predictors. J Cardiovasc Electrophysiol, 2021.

  1. The authors mention Bonferroni correction, although in the statistical analysis and tables there is no mention of the Bonferroni-corrected p value.

We removed the information about the Bonferroni correction from “Statistical analysis” section.

Reviewer 2 Report

The authors reported the findings of patients undergoing transesophageal echocardiogram before cardioversion/ablation in patients in AF/flutter. A total of 3109 patients were enrolled in the so-called LATTEE registry. Prevalence of left appendage thrombus was assessed. Interestingly, patients on VKA showed higher incidence of LAT than those of NOAC even after adjusting for potential confounders. Among those receiving NACOS, those on apixaban showed higher incidence of LAT than those on rivaroxaban or dabigatran However, after adjusting for confounders, those differences disappeared. 

Main comments:

Paper is interesting and well-written. Limitations of this registry are well acknowledged by the authors. Data on low-dose NOAC are of interest as they are frequently used in clinical practice. The authors should report specific low doses given to patients of apixaban (2,5?); rivaroxaban (10? vs 15?) and dabigatran (110?). Besides, % of LAT in patients under low-dose NOAC should also be reported or alternatively expressed as Figure 2 D. Rates of low-dose rivaroxaban and low-dose apixaban are probably pretty similar I guess (1,4% per group?)

Author Response

Reviewer 2

Comments and Suggestions for Authors

The authors reported the findings of patients undergoing transesophageal echocardiogram before cardioversion/ablation in patients in AF/flutter. A total of 3109 patients were enrolled in the so-called LATTEE registry. Prevalence of left appendage thrombus was assessed. Interestingly, patients on VKA showed higher incidence of LAT than those of NOAC even after adjusting for potential confounders. Among those receiving NACOS, those on apixaban showed higher incidence of LAT than those on rivaroxaban or dabigatran However, after adjusting for confounders, those differences disappeared. 

Main comments:

Paper is interesting and well-written. Limitations of this registry are well acknowledged by the authors. Data on low-dose NOAC are of interest as they are frequently used in clinical practice.

The authors should report specific low doses given to patients of apixaban (2,5?); rivaroxaban (10? vs 15?) and dabigatran (110?). Besides, % of LAT in patients under low-dose NOAC should also be reported or alternatively expressed as Figure 2 D. Rates of low-dose rivaroxaban and low-dose apixaban are probably pretty similar I guess (1,4% per group?)

We included this information as follows:

“Reduced doses for rivaroxaban, dabigatran and apixaban were 15mg o.d., 110mg b.i.d. and 2.5mg b.i.d., respectively. Only one patient in the rivaroxaban group received a reduced dose of 10mg o.d..” (page 5, lines 161-162)

Rates of left atrial thrombus for low-dose rivaroxaban and low-dose apixaban were 14% and 29%, respectively, and the difference was statistically significant, as shown in Figure 1B.

Round 2

Reviewer 2 Report

Thank you for your answers. I do not have further issues.